# Aggregation mechanism of colloidal kaolinite in aqueous solutions with electrolyte and surfactants

Yang Hu[1], Qingyang Yang[1], Jue Kou[1], Chunbao Sun[1], Hongliang Li[2]*

1 School of Civil and Resources Engineering, University of Science and Technology Beijing, Beijing, China,
2 College of Mining Engineering, Taiyuan University of Technology, Taiyuan, China

* lihongliang222@126.com

**Data Availability Statement:** All relevant data are within the manuscript and its Supporting Information files.

**Funding:** National Natural Science Foundation of China (project No. 51804021 and No. 51804213)

## Abstract

In this study, the effect of surfactants and electrolytes on stability of kaolinite dispersions was analyzed by measuring suspension transmittance, zeta potential, and adsorption. It was experimentally found that the compression of kaolinite electric double layer caused by NaCl addition may reduce the electrostatic repulse force to facilitate the aggregation of kaolinite particles. Surfactant facilitate the aggregation of kaolinite particles mainly through the adsorption of it on the surface of kaolinite to generate hydrophobic force. Compared to anionic surfactant, the cationic surfactant has a better flocculation effect because it can be used in a wide pH range and its adsorption can reduce the electrostatic repulse force between kaolinite particles.

## 1. Introduction

The layered structure of kaolinite is composed of a layered silicon–oxygen tetrahedron and a layered aluminum–oxygen octahedron [1, 2]. This special structure provides it with very good hydrophilicity [3, 4]. In general, mineral tailings contain a large amount of fine clay minerals such as kaolinite. The size of these kaolinite particles can reach a colloidal level after milling and grinding during mineral processing. Weak gravity coupled with the electrostatic repulsion and hydration repulsion between the kaolinite particles is not enough to deposit them quickly, leading to their high stability in water [5–7]. However, considering the environmental protection, most of the waste tailings produced during mineral processing have to be dewatered for reuse [8–10]. Because the high stability of kaolinite suspension may impede the solid–liquid separation, a pre-aggregation of colloidal kaolinite particles is necessary to ensure an acceptable dewatering efficiency.

According to the literature, the main methods to aggregate colloidal kaolinite particles are the addition of electrolytes and surfactants to change the surface properties such as charge and hydrophilicity of kaolinite particles, or combining these particles with polymer molecules [11–13]. For example, cationic polyacrylamide and cetyl trimethyl ammonium bromide may flocculate kaolinite particles through charge neutralization and sweeping mechanism. The presence of inert electrolyte ions such as Na$^+$ and K$^+$ in kaolinite suspension may compress the

and China Scholarships Council (CSC) under
project No. 201906935041.

**Competing interests:** The authors have declared
that no competing interests exist.

electric double layer of kaolinite particles, leading to electrolyte agglomeration [14, 15]. Generally, kaolinite suspension can be coagulated easily when the concentration of NaCl or KCl is greater than 7 mmol/L [16].

The aggregation effects of the electrolyte and polymer flocculation methods are good, but the electrolyte flocculation method is not suitable for solutions with high pH, and the polymer flocculants are generally more expensive compared to the other two methods. Furthermore, the strong hydration repulsion force derived from the thick hydrated film around the hydrophilic surface of kaolinite may also weaken the flocculation effect of the electrolyte and polymer flocculation methods. In this case, surfactants such as sodium dodecyl sulfate and sodium dodecyl benzenesulfonate are a better choice because they aggregate kaolinite particles by generating a strong hydrophobic attractive force between them, which is be affected by pH, and the hydration shell may break due to the hydrophobization of the particles [17–20]. To study the induction of surfactants on the hydrophobic flocculation of kaolinite particles, it is necessary to understand the generation and variation of the hydrophobic force between kaolinite particles.

In this study, we analyzed the flocculation behavior of kaolinite in a solution of dodecylamine chloride (DDACl) and sodium oleate (SO). The sedimentation of kaolinite in electrolyte and surfactant solution was recorded using an ultraviolet spectrophotometer. The zeta potential and adsorption density of surfactant were measured to study the aggregation mechanism of kaolinite particles. The variation of forces between kaolinite particles was calculated using the expanded DLVO theory to further explain the aggregation mechanism. The objective was to learn more about the agglomeration mechanism of colloidal kaolinite induced by electrolyte and surfactant.

## 2. Experimental

### 2.1 Materials

Kaolinite was purchased from Jinyan Kaolinite Company (China), and the particle size of the sample was less than 2.00 μm. DDACl, SO, NaCl, NaOH, and HCl were of analytical purity grade and supplied by Sinopharm Chemical Reagent Co. Ltd (China). In this work, water was purified using the Barnstead EASYpure II system to decrease residual conductivity to less than 1 μS/cm.

### 2.2 Sedimentation experiments

To fully disperse kaolinite particle in water, 200 mL 0.2 wt% of kaolinite aqueous suspension was stirred at 6000 rpm for 5 min, and then shaken in an ultrasonic cleaner for 5 min. Next, the pH of the slurry was adjusted and a certain dosage of surfactant (DDACl or SO) and/or NaCl was added while stirring at 400 rev/min for 15 min. Subsequently, 5 mL of suspension was moved into a quartz cell to settling for 1 h. Meanwhile, the light transmittance of suspension was measured using an AquaMate Plus spectrophotometer (United States) at the wavelength of 600 nm. The remaining kaolinite suspension was centrifuged at 4000 rpm for 10 min to separate solid and liquid. The solid was dried to measure the contact angle using the Washburn method. The liquid was measured with spectrophotometer to obtain the final concentration of surfactant which was used to calculate the adsorption density ($q$) of surfactant with following equation:

$$q = \frac{V_o(C_o - C)}{mA_{sp}}.$$  (1)

Here, $V_o$ is the volume of kaolinite suspension, $C_o$ and $C$ are the initial and residual concentration of surfactant, respectively, $m$ is the mass of kaolinite sample, and $A_{sp}$ is the specific surface area of the kaolinite particles, $A_{sp} = 11.67$ m²/g in this work.

## 2.3 Characterization

A Quantachrome Autosorb-IQ gas adsorption analyzer (United States) was used to measure specific surface area of the kaolinite powders based on the Brunauer–Emmett–Teller (BET) method. To obtain the zeta potential of kaolinite particles, a certain amount of surfactant or NaCl was added in 250 mL 2 wt.% kaolinite suspension and then the suspension was stirred at 300 rev/min for 15 min. The zeta potential of kaolinite under different conditions was measured with a colloidal dynamics ZetaProbe analyzer (United States) based on electroacoustic technology. Subsequently, the measurement was carried out at 25±2°C and the pH was adjusted automatically with 0.1 mol/l of HCl and 0.1 mol/l of NaOH. The Washburn method based on literature was used to calculate the contact angle of kaolinite powder before and after surfactant adsorption [21].

## 2.4 Calculation using extended DLVO theory

The total interaction energy between two interfaces $V_T$ can be expressed as follows [22, 23]:

$$V_T = V_R + V_A + V_H \tag{2}$$

The electrostatic interaction energy between kaolinite particles is expressed as follows:

$$V_R = \frac{8a\varepsilon(k_B T)^2}{e^2 Z^2}\tanh^2(\frac{Ze\psi}{4k_B T})\exp(-kh) \tag{3}$$

Here, $\alpha$ (m) is the average radius of kaolin particles, $\varepsilon$ is the dielectric constant of the dispersion medium (the dielectric constant of water at 25°C is $6.95\times10^{-10}$ F/m); $k\_B$ is the Boltzmann constant ($k\_B = 1.38 \times 10^{-23}$ J/K); $T$ (K) is the absolute temperature; $e$ is the electric charge ($e = 1.602 \times 10^{-19}$ C); $Z$ is the valence of ions in the solution; $\Psi$ (V) is the absolute value of Zeta potential on the particle surface; $h$ (m) is the closest separation distance between two particles; and $k^{-1}$ is the Debye length that represents the thickness of the electric double layer. The Debye length is expressed as follows:

$$k = \left[(1000e^2 N_A/\varepsilon k_B T)\sum_i z_i^2 M_i\right]^{1/2} \tag{4}$$

where $N_A$ is the Avogadro constant ($N_A = 6.022 \times 1023$ mol⁻¹), $Z_i$ is the valence of ion $i$, and $M_i$ (mol/L) is the molar concentration of ion $i$. For the energy calculation of kaolinite without surfactants or electrolytes, 0.2 mmol/L of NaCl was added to supply ion $i$.

The van der Waals interaction energy between particles is expressed as follows:

$$V_A = -\frac{A_{131}a}{12h} \tag{5}$$

where $A_{131}$ (J) is the Hamaker constant of kaolinite in water, which can be calculated using the following formula:

$$A_{131} \approx (\sqrt{A_{11}} - \sqrt{A_{33}})^2 \tag{6}$$

Here, $A_{11}$ ($14.89\times10^{-20}$J) and $A_{33}$ ($4.4\times10^{-20}$J) [24] are the Hamaker constants for kaolinite and water in a vacuum, respectively.

The hydrophobic interaction energy of kaolinite is expressed as follows [25]:

$$V_H = -2.51 \times 10^{-3} a k_1 h_0 \exp(-h/h_0) \tag{7}$$

$$k_1 = \frac{\exp(\theta/100) - 1}{e - 1} \tag{8}$$

$$h_0 = 11.2 \times 10^{-9} k_1 \tag{9}$$

where $e$ is the base of natural logarithm ($e \approx 2.7182...$) and $\theta$ (˚) is the contact angle of water on the surface of kaolinite.

## 3. Results and discussion

Fig 1 shows the light transmittance of 0.2 wt% of kaolinite suspension as a function of settling time at the pH values of 3.04, 6.98, and 10.01. It can be observed that light transmittance increases from 0% to approximately 55% within 1 h when the suspension is acidic, implying that kaolinite particles settle quickly. When the pH value increases to neutral and alkaline, the light transmittance shows no change, indicating that kaolinite suspension is very stable. This phenomenon indicated that the aggregation of kaolinite particles occurred mainly under acidic condition, but not under neutral and alkaline conditions due to the variation in the charges of kaolinite surfaces when pH changed.

As presented in Fig 2, the zeta potential of kaolinite decreases from 30.51 mV to –174.44 mV when pH increases from 2.02 to 9.98. The variation in zeta potential may be attributed to the variation of charges on the edge surfaces (E-faces) and the aluminium-oxygen octahedron surfaces (O-faces). The chemical structures of E-face, O-face and silicon-oxygen tetrahedron surface (T-face) were presented in Fig 3. It is known that the charges on E-face and O-face are pH-dependent because of the chemical reactions shown in formulas (10), (11) and (12), while T-face always remain negative because of isomorphic substitution ($Si^{4+}$ replaced by $Al^{3+}$ or $Fe^{3+}$). When pH value was low, the charges on E-face and O-face will be positive while T-face

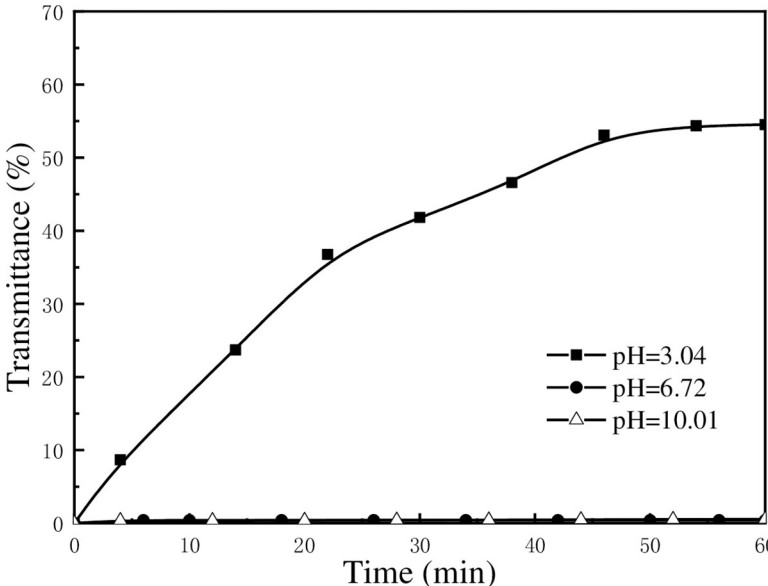

**Fig 1. Light transmittance of kaolinite suspension as a function of settling time.**

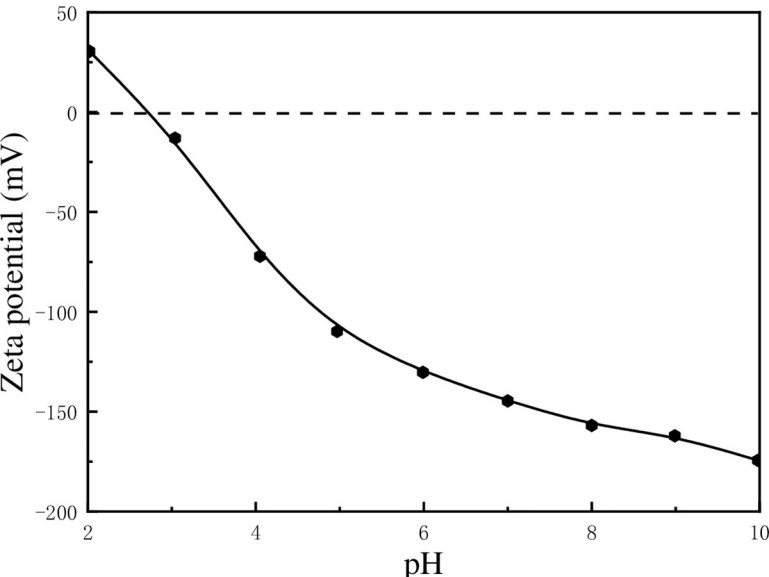

**Fig 2. Zeta potential of kaolinite particles as a function of pH in aqueous solution.**

is still negatively charged, leading to the F–E (face-edge) and F–F (face-face) aggregations of kaolinite particles as shown in Fig 4. On the other hand, kaolinite particles also may be aggregated with E-E (edge-edge) form by the van der Waals interaction between them if the absolute value of charge on E-face is very small.

$$Al-OH + H^+ \Leftrightarrow Al-OH_2^+ \tag{10}$$

$$Al-OH + OH^- \Leftrightarrow Al-O^- \tag{11}$$

$$Si-OH + OH^- \Leftrightarrow Si-O^- \tag{12}$$

Fig 5 shows the effect of NaCl concentration on the settling performance of kaolinite particles at pH = 5. The increase rate of light transmittance is small when NaCl concentration is low, whereas it increased rapidly when NaCl concentration increases. When NaCl concentration reaches 10 mmol/L, the increase rate of transmittance changes slightly. As shown in the

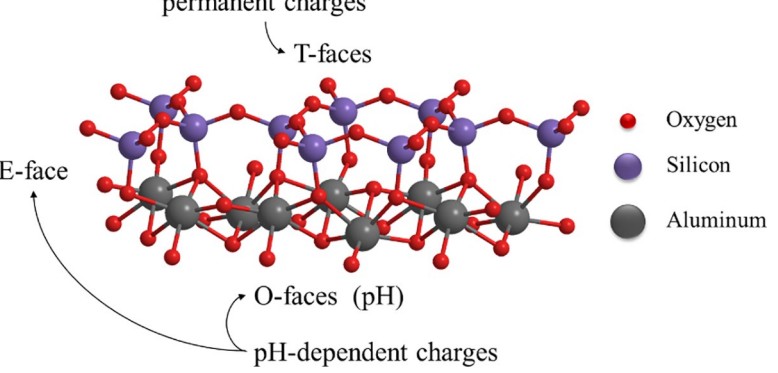

**Fig 3. The chemical structure of E-face, O-face and T-face of kaolinite.**

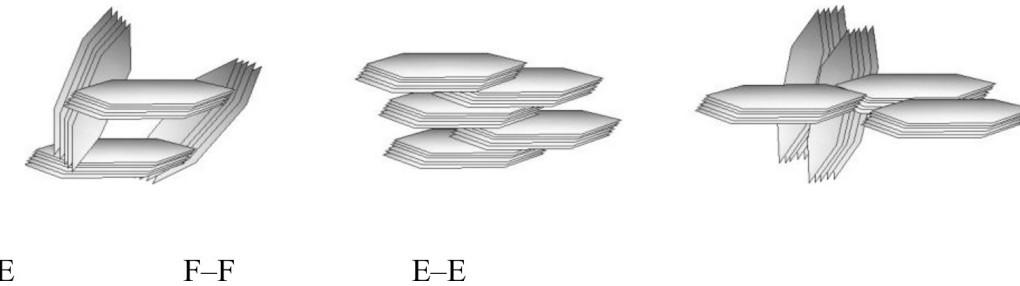

F–E          F–F          E–E

**Fig 4. Schematic diagrams of agglomeration forms of kaolinite particles.**

figure, the transmittance curves at 10 mmol/L, 15 mmol/L, and 20 mmol/L of NaCl almost coincide. The variation in transmittance caused by NaCl may be attributed to the changes in zeta potential, as shown in Fig 6. It is known that Na$^+$ is an inert ion, and its influence on the stability of kaolinite suspension has been widely reported in recent years. Like other common inert ions, Na$^+$ can compress the electric double layer on the surface of kaolinite to decease zeta potential. The $V_T$ was calculated with classical DLVO theory ($V_T = V_R + V_A$) and the results were shown in Fig 7, the weakening of zeta potential resulted in the reduction of total potential energy between kaolinite particles. It is apparent that the increase in NaCl concentration reduces the energy barrier which is the obstacle for aggregating kaolinite particles. It can be observed from Fig 7 that as the NaCl concentration increases, the energy barrier between particles decreases, which is in consistent with the conclusion that increasing the Na$^+$ content in the suspension is beneficial for the aggregation of kaolinite particles.

Besides electrolytes, surfactants are also good flocculants used to aggregate clay minerals. Fig 8 shows the settlement of kaolinite particles with or without surfactant solution under acid, neutral, and alkaline conditions. It can be observed that the transmittance of kaolinite-DDACl suspension under neutral and alkaline conditions is higher and increases rapidly than that

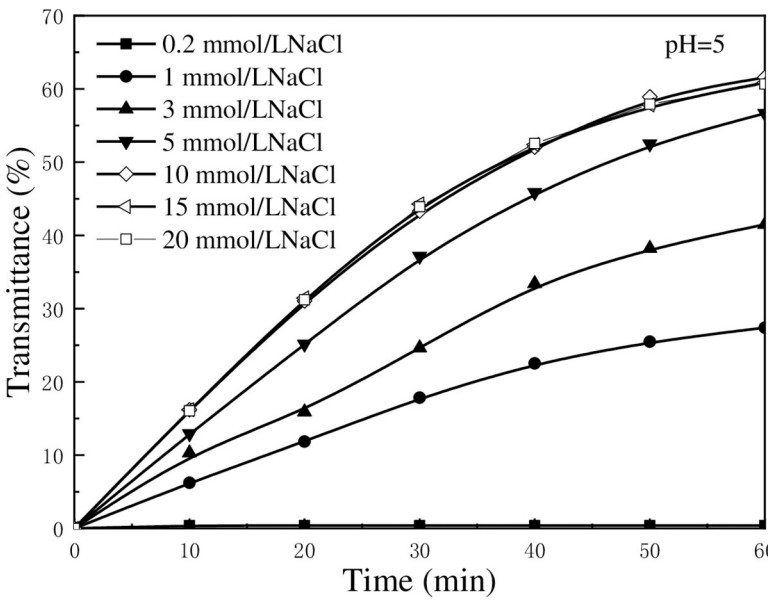

**Fig 5. Light transmittance of kaolinite suspension in different concentrations of NaCl solution as a function of settling time.**

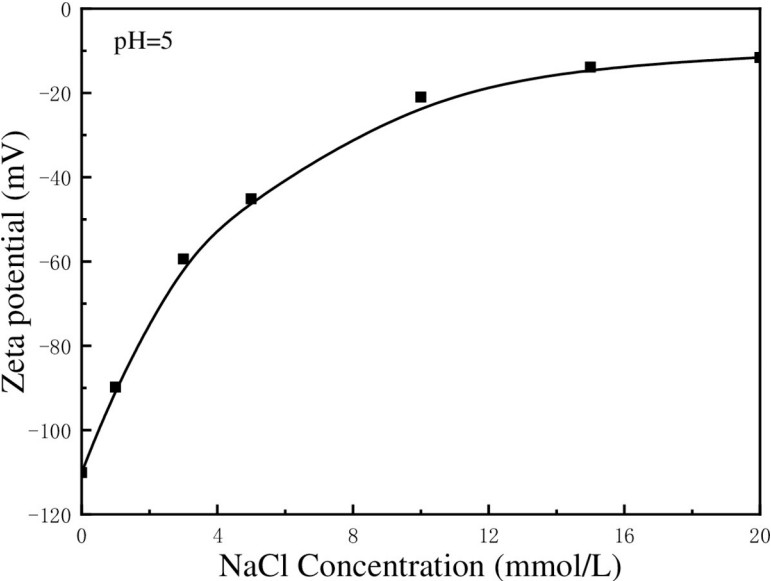

**Fig 6. Zeta potential of kaolinite particles at different NaCl concentration.**

under acid condition, indicating that the flocculation effect of DDACl on kaolinite particles was better in neutral and alkaline conditions compared to acid condition. On the contrary, kaolinite particles prefer to flocculate under acid condition, but not under neutral and alkaline conditions when SO was added.

To understand the reason for different effects of DDACl and SO on the flocculation of kaolinite particles under various pH conditions, the zeta potentials of kaolinite-DDACl and kaolinite–SO were measured and showed in Fig 9. It was observed that the zeta potential of kaolinite particles increases at pH = 2–10 after DDACl is added, which might be attributed to the electrical neutralization on the surface of kaolinite particles when positively charged

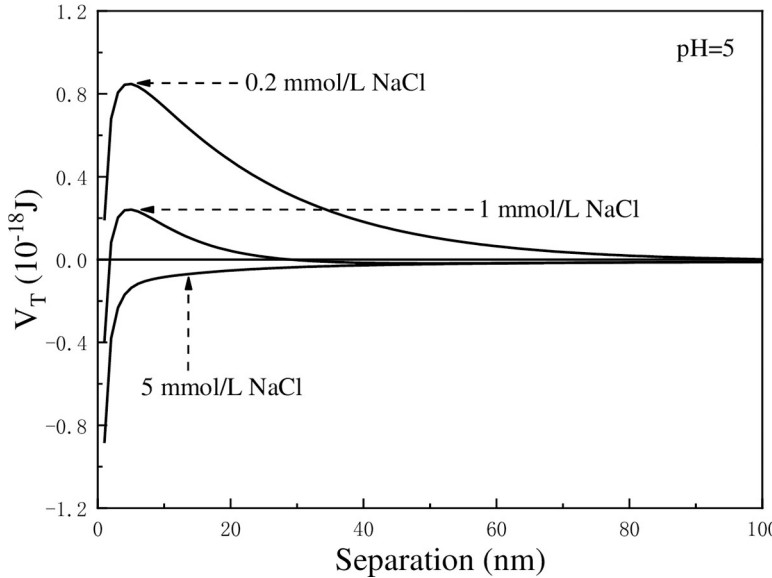

**Fig 7. The effect of NaCl concentrations on the total potential energy between kaolinite particles.**

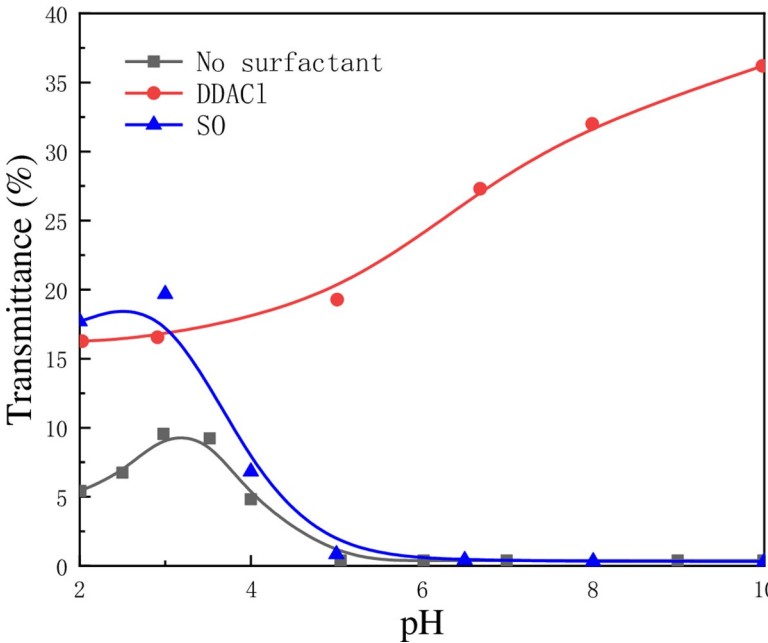

**Fig 8. Light transmittance of kaolinite suspension with or without surfactants (concentrations of both surfactants was $5\times10^{-4}$ mol/L, and settling time was 5 min) as function of pH.**

DDACl was adsorbed on the surface. For kaolinite-SO, the zeta potential decreases at pH = 2–7 while maintaining nearly no change at pH = 7–10, which might be due to the adsorption of negatively charged SO on the positively charged edge surface and hydroxyl-ter-minated planes. To understand the effect of zeta potential on the adsorption of DDACl and SO more clearly, their adsorption densities at different pH values were calculated, as shown in

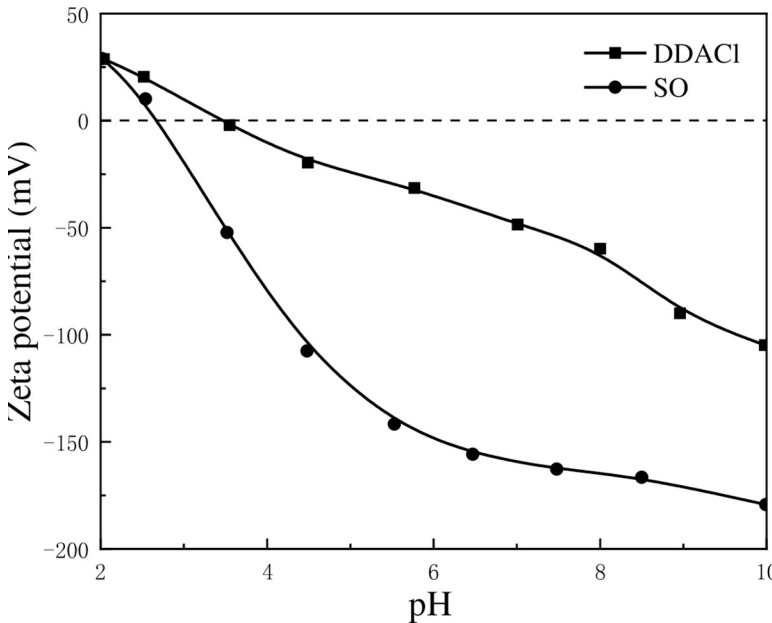

**Fig 9. Zeta potential of kaolinite particles in $5\times10^{-4}$ mol/L of DDACl and SO at different pH values.**

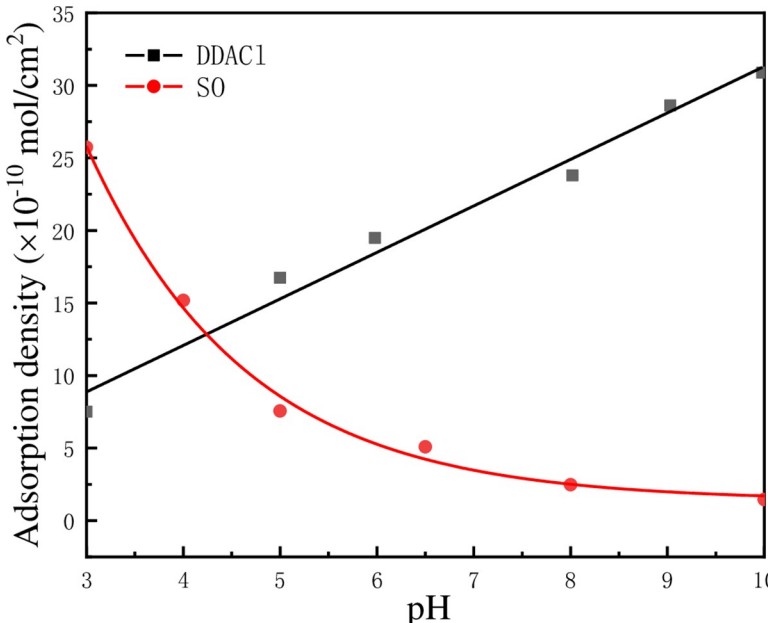

**Fig 10. Adsorption densities of DDACl and SO on surface of kaolinite particles (initial concentration of both surfactants was 5×10⁻⁴ mol/L) as function of pH.**

Fig 10. It can be observed that the fitted adsorption density curves of DDACl and SO are very different. The curve of DDACl increases linearly, whereas the curve of SO shows an exponentially downward trend. It is known that the kaolinite surface may turn to hydrophobic after surfactants was adsorbed on it, which may facilitate the aggregation of kaolinite particles. As the surfactants adsorption was carried out through electrostatic attraction, DDACl flocculated kaolinite appropriately at a high pH, and SO flocculated kaolinite at a low pH, which corresponded well with the results shown in Fig 8.

According to the above discussion, the electrolyte in kaolinite suspension facilitates the aggregation of kaolinite particles by decreasing the energy barrier between kaolinite particles, which can be explained using the classical DLVO theory. Similarly, the surfactants also reduce the energy barrier to make the aggregation of kaolinite particles easier. However, due to the generation of hydrophobic force between kaolinite particles, the classical DLVO theory could not be used to explain the surfactant-induced flocculation. To calculate the total potential energy between two kaolinite particles in surfactant solution, Eq (7) was used in which the hydrophobic force was considered. According to this formula, the hydrophobic force should be proportional to the contact angle of kaolinite particle. The contact angles of kaolinite particles adsorbed on DDACl and SO (initial concentration of both surfactants was 5×10⁻⁴ mol/L, pH = 4, and adsorbing time was 15 min) were obtained using the Washburn method, and the values were 89.62˚ and 88.03˚, respectively. The contact angle of kaolinite–SO was slightly greater than that of kaolinite-DDACl because the adsorbing capacity of SO was more than that of DDACl, as shown in Fig 10.

When pH is 4, the zeta potentials of kaolinite, kaolinite–DDACl, and kaolinite–SO were –66.41, –10.01, and –80.14 mV, respectively. Although the edge and hydroxyl-terminated plane of a kaolinite particle started to charge positively at this pH, the entire kaolinite particle was still negatively charged. In this case, the electrostatic interaction force between kaolinite particles was treated as the repulsive force when the electrostatic interaction energy and total potential energy were calculated. As shown in Fig 11, the electrostatic interaction energy of

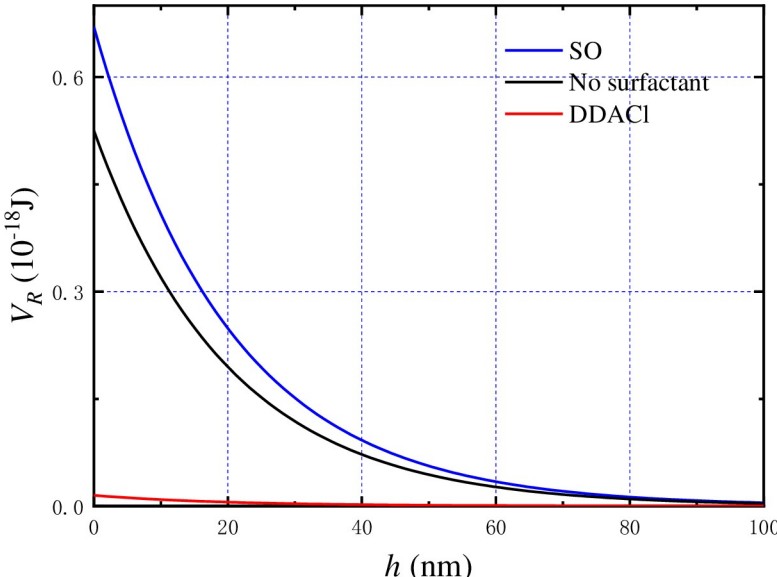

**Fig 11. Electrostatic interaction energy of kaolinite particles with or without surfactants as function of particle spacing.**

kaolinite-SO is greater than that of kaolinite without surfactant, and much greater than that of kaolinite–DDACl, which corresponds well with the zeta potential results. After the hydrophobic force and the van der Waals force were considered, the total potential energy was calculated as follows: kaolinite without surfactant > kaolinite–SO > kaolinite–DDACl, as shown in Fig 12 ($V_T = V_R + V_A + V_H$). It was apparent that there was no energy barrier for kaolinite–DDACl particles, implying that they can be flocculated very easily. The energy barrier of kaolinite–SO is smaller than that of kaolinite without surfactant, indicating that kaolinite–SO particles can aggregate more easily than pure kaolinite particles, which corresponds well with the results

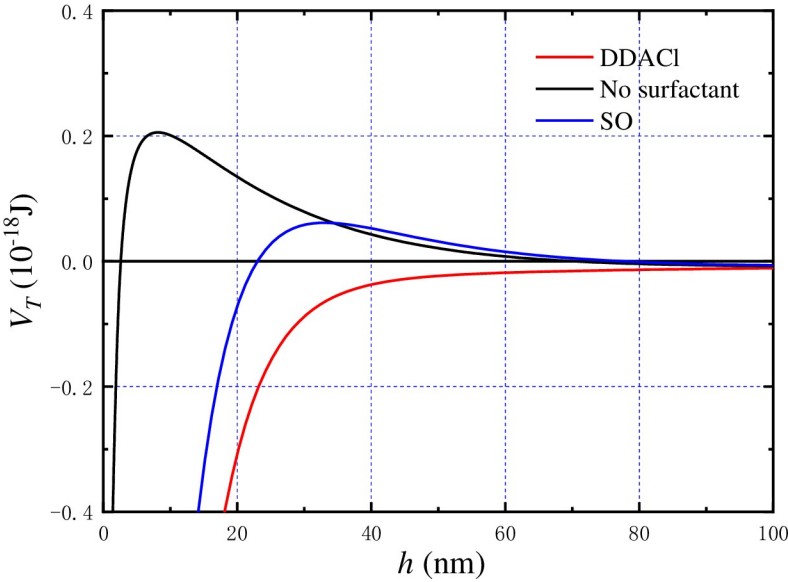

**Fig 12. Total potential energy of kaolinite particles with or without surfactants as function of particle spacing.**

shown in Fig 8. In particular, it was concluded that DDACl was a better flocculent than SO without considering strong acidic conditions.

## 4. Conclusions

(1) It was experimentally found that the surface potential of kaolinite particles shifts from being negatively to positively charged with decreasing pH, leading to a kaolinite suspension that remains stable under acidic condition and aggregates under neutral and alkaline conditions. The effect of NaCl on the stability of the colloidal kaolinite suspension was mainly due to the compression of the electric double layer around the particles, which reduced the zeta potential on the surface of the particles and the electrostatic repulsion between the particles.

(2) When surfactants were added to the kaolinite suspension, the generation of hydrophobic forces significantly reduced the energy barrier and facilitated the particle aggregation. Compared to anionic surfactant, the cationic surfactant has a better flocculation effect because it can be used in a wide pH range and its adsorption can reduce the electrostatic repulse force between kaolinite particles.

## Supporting information

**S1 File.**
(DOCX)

## Author Contributions

**Data curation:** Yang Hu, Qingyang Yang.

**Formal analysis:** Jue Kou, Hongliang Li.

**Supervision:** Chunbao Sun.

**Writing – original draft:** Yang Hu.

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
