## [Decision Letter · Decision Letter 0]

29 Jun 2020

PONE-D-20-16933

Aggregation mechanism of colloidal kaolinite in aqueous solutions with electrolyte and surfactants

PLOS ONE

Dear Dr. Li,

Thank you for submitting your manuscript to PLOS ONE. After careful consideration, we feel that it has merit but does not fully meet PLOS ONE’s publication criteria as it currently stands. Therefore, we invite you to submit a revised version of the manuscript that addresses the points raised during the review process.

We look forward to receiving your revised manuscript.

Kind regards,

Jie Zheng, Ph.D

Academic Editor

PLOS ONE

Journal Requirements:

'This research was financial supported by the National Natural Science Foundation of China (project No. 51804021 and No. 51804213) and China Scholarships Council (CSC) under project No. 201906935041.'

'The author(s) received no specific funding for this work.'

Reviewers' comments:

Reviewer's Responses to Questions

**Comments to the Author**

1. Is the manuscript technically sound, and do the data support the conclusions?

Reviewer #1: Yes

Reviewer #2: Yes

2. Has the statistical analysis been performed appropriately and rigorously? 

Reviewer #1: Yes

Reviewer #2: N/A

3. Have the authors made all data underlying the findings in their manuscript fully available?

Reviewer #1: Yes

Reviewer #2: Yes

4. Is the manuscript presented in an intelligible fashion and written in standard English?

Reviewer #1: Yes

Reviewer #2: Yes

5. Review Comments to the Author

Reviewer #1: The study focused on the electrolytes and surfactants facilitated kaolinite aggregation, from the perspective of electrostatic and hydrophobic interactions. Although it provided a series of analysis to characterize the aggregation process, there still exist several issues need to be further addressed in the paper, and a minor revision is recommended.

1. Firstly, the language and writing need to be improved. There even exist several incomplete sentences in the paper, e.g. Section 2.3

2. The study explained the pH dependent kaolinite suspension aggregation process by three aggregation models (Fig. 3). But there is no E/ F-face information introduced in the main text. Please provide more detailed introduction to the edge surfaces and planes of kaolinite to support the models.

3. In surfactants-kaolinite system, kaolinite-DDACl suspension under neutral and alkaline conditions exhibited facilitated aggregation. To explain the phenomenon, the authors claimed that “the aggregation of kaolinite particles facilitated by the increase of negative charges on the kaolinite surface”, in the contrary, the author later discussed on the electrical neutralization on the surface of kaolinite particles when positively charged DDACl was adsorbed on the surface. Please provide a solid explanation on the surfactants facilitated aggregation process.

Reviewer #2: The author studied the different aggregation mechanisms of kaolinite suspension at different pH, NaCl concentration, or in the presence of different surfactants. The topic is interesting and worth studying, however, the manuscript need improvement before further consideration for publication. My comments are listed below.

1. Both page and line numbers should be added for manuscript review.

2. Equations (2), (6), (7) and (8) should be checked.

3. For each experimental data in this manuscript, how many measurements were done?

4. Transmittance. What is the quantitative correlation between transmittance and remaining kaolinite in water?

5. The highest transmittance was only 60-65% in this study. From a practical point, how much settling (in %) is desired for pre-aggregation?

6. The author may consider adding a Kaolinite structure picture to indicate different E-, O- and T- faces. Also please mention what kind of isomorphic substitution. Si4+ replaced by Al3+ or Fe3+ and Al3+ replaced by Ca2+ and Mg2+?

7. The author should provide a table for VR, VA, VH and VT values.

8. Figure 6 and Figure 11. Were VT calculated using classical DLVO theory (Vt=VR+VA) or extended DLVO theory (VT=VR+VA+VH)? Please note.

9. What is the contact angle for kaolinite without surfactant?

6. PLOS authors have the option to publish the peer review history of their article (what does this mean?). If published, this will include your full peer review and any attached files.

Reviewer #1: No

Reviewer #2: No

---

## [Author Response · Author response to Decision Letter 0]

11 Aug 2020

Response to Reviewer

Reviewer #1: 

The study focused on the electrolytes and surfactants facilitated kaolinite aggregation, from the perspective of electrostatic and hydrophobic interactions. Although it provided a series of analysis to characterize the aggregation process, there still exist several issues need to be further addressed in the paper, and a minor revision is recommended.

Point 1: Firstly, the language and writing need to be improved. There even exist several incomplete sentences in the paper, e.g. Section 2.3

Response: The language of this paper has been polished by Editage company, and the incomplete sentence in Section 2.3 has been finished and marked in red.

Point 2: The study explained the pH dependent kaolinite suspension aggregation process by three aggregation models (Fig. 3). But there is no E/ F-face information introduced in the main text. Please provide more detailed introduction to the edge surfaces and planes of kaolinite to support the models.

Response: The introduction for to the edge surfaces and planes of kaolinite were rewrote and marked in red (line 143-157). The chemical structure was shown in Fig. 3 to explain the difference of E-, O- and T-faces. 

Point 3: In surfactants-kaolinite system, kaolinite-DDACl suspension under neutral and alkaline conditions exhibited facilitated aggregation. To explain the phenomenon, the authors claimed that “the aggregation of kaolinite particles facilitated by the increase of negative charges on the kaolinite surface”, in the contrary, the author later discussed on the electrical neutralization on the surface of kaolinite particles when positively charged DDACl was adsorbed on the surface. Please provide a solid explanation on the surfactants facilitated aggregation process.

Response: The meaning of sentence “the aggregation of kaolinite particles facilitated by the increase of negative charges on the kaolinite surface” was that the increase of negative charges on the kaolinite surface may help to absorb more positively charged DDACl, and then the aggregation process was facilitated. However, the results in Fig. 8 and the corresponding describes “It can be observed that the transmittance of kaolinite-DDACl suspension under neutral and alkaline conditions is higher and increases rapidly than that under acid condition” can’t support this conclusion (The similar conclusion was given after the discussion of Fig. 9 and 10). In this case, the sentence “indicating that the aggregation of kaolinite particles facilitated by the increase of negative charges on the kaolinite surface” was changed to “indicating that the flocculation effect of DDACl on kaolinite particles was better in neutral and alkaline conditions compared to acid condition” (line 179-181).

Reviewer #2: 

The author studied the different aggregation mechanisms of kaolinite suspension at different pH, NaCl concentration, or in the presence of different surfactants. The topic is interesting and worth studying, however, the manuscript need improvement before further consideration for publication. My comments are listed below.

Point 1: Both page and line numbers should be added for manuscript review.

Response: The page and line numbers were added in the revised manuscript.

Point 2: Equations (2), (6), (7) and (8) should be checked.

Response: Equations (2), (6), (7) and (8) were checked according to book “Interfacial Separation of Particles” and “Principles of colloid and surface chemistry”, the corresponding literatures were reference [22] and [23].

Point 3: For each experimental data in this manuscript, how many measurements were done?

Response: In this work, each experimental data was measure three times and the average value of them was used to draw the curves.

Point 4: Transmittance. What is the quantitative correlation between transmittance and remaining kaolinite in water?

Response: The higher the Transmittance, the less the remaining kaolinite in water.

Point 5: The highest transmittance was only 60-65% in this study. From a practical point, how much settling (in %) is desired for pre-aggregation?

Response: The main purpose of this paper was trying to study the aggregation mechanism of colloidal kaolinite in aqueous solutions with electrolyte and surfactants, so the longest settling time was set as 1 h and the transmittance didn’t reach the maximum value. Besides, the addition of surfactant may accelerate the aggregation process. For example, the transmittance of kaolinite-DDACl was about 37% when settling time was 5 min while the transmittance of kaolinite without surfactant was just 10%. In practical production, the surfactants were added in pre-aggregation stage, and then the thickner will continue to settle the particles until the solid content in its supernatant reach the requirement for production.

Point 6: The author may consider adding a Kaolinite structure picture to indicate different E-, O- and T- faces. Also please mention what kind of isomorphic substitution. Si4+ replaced by Al3+ or Fe3+ and Al3+ replaced by Ca2+ and Mg2+?

Response: The Kaolinite structure picture was added in Fig. 3. The isomorphic substitution in T-face should be Si4+ replaced by Al3+ or Fe3+, the corresponded explain was added in line 148 and 149.

Point 7: The author should provide a table for VR, VA, VH and VT values.

Response: The VR, VA, VH and VT values were changing with the increase of separation between kaolinite particles as shown in fig. 7 and 12, it was hard to put all the values in a table.

Point 8: Figure 6 and Figure 11. Were VT calculated using classical DLVO theory (Vt=VR+VA) or extended DLVO theory (VT=VR+VA+VH)? Please note.

Response: The VT in Figure 6 (fig. 7 in revised manuscript) was calculated with classical DLVO theory and VT in Figure 11 (fig. 12 in revised manuscript) was calculated with extended DLVO theory. they were noted in line 168 and 229.

Point 9: What is the contact angle for kaolinite without surfactant?

Response: The contact angle for kaolinite in deionized water was 0°.

---

## [Editor Report · Decision Letter 1]

17 Aug 2020

Aggregation mechanism of colloidal kaolinite in aqueous solutions with electrolyte and surfactants

PONE-D-20-16933R1

Dear Dr. Li,

We’re pleased to inform you that your manuscript has been judged scientifically suitable for publication and will be formally accepted for publication once it meets all outstanding technical requirements.

Kind regards,

Jie Zheng, Ph.D

Academic Editor

PLOS ONE
---

## [Editor Report · Acceptance letter]

10 Sep 2020

PONE-D-20-16933R1 

Aggregation mechanism of colloidal kaolinite in aqueous solutions with electrolyte and surfactants 

Dear Dr. Li:

I'm pleased to inform you that your manuscript has been deemed suitable for publication in PLOS ONE. Congratulations! Your manuscript is now with our production department. 

Kind regards, 

on behalf of

Dr. Jie Zheng 

Academic Editor

PLOS ONE